# Is the Prediction Set Size Well-Calibrated? A Closer Look at Uncertainty in Conformal Prediction

## Abstract

Given its flexibility and low computation, conformal prediction (CP) has become one of the most popular uncertainty quantification methods in recent years. In deep classifiers, CP will generate a prediction set for a test sample that satisfies the $(1 - \alpha)$ coverage guarantee. The prediction set size (PSS) is then considered a reflection of the predictive uncertainty. However, it is unknown whether the predictive uncertainty of CP is aligned with its predictive correctness, which is an imperative property for predictive uncertainty. This work answers this open question by investigating the uncertainty calibration of CP in deep classifiers. We first give a definition for the uncertainty calibration of CP by building a connection between PSS and prediction accuracy and then propose a calibration target for CP based on a theoretical analysis of the predictive distributions. Given this defined CP calibration, we present an empirical study on several classification datasets and reveal their weak calibration of CP. To strengthen the calibration of CP, we propose CP-aware calibration (CPAC), a bi-level optimization algorithm, and demonstrate the effectiveness of CPAC on several standard classification datasets by testing models including ResNet, Vision Transformer and GPT-2.

## 1 Introduction

Due to its low computational overhead and distribution-free assumption, conformal prediction (CP) Shafer & Vovk (2008); Romano et al. (2020); Angelopoulos et al. (2021) has become a dominant approach to uncertainty quantification (UQ). CP has been successfully adopted in various machine-learning applications, including object detection Timans et al. (2024), pose estimation Yang & Pavone (2023), pixel-level image understanding Brunekreef et al. (2024) and natural language understanding Quach et al. (2024); Gui et al. (2024); Mohri & Hashimoto (2024). The predictive uncertainty from CP stems from the *frequentist approach* to uncertainty, i.e., producing a confidence interval will contain the true value with a specified probability (e.g., 90% or 95%). In a classification task, CP will produce a prediction set $\mathcal{S}$ for a test sample that is theoretically guaranteed to contain the ground-truth class label with a high probability (e.g., 90%). Although it is desirable to have a prediction set with a coverage guarantee, as a probabilistic forecast model, it is also important that the uncertainty is consistent with the decision's reliability Mincer & Zarnowitz (1969); Kochenderfer (2015), known as *calibration* Zadrozny & Elkan (2001); Gneiting et al. (2007).

For a multi-class classification model, the confidence score of the predicted label from a predictive distribution has traditionally been used as a measure of uncertainty. In that case, model calibration aims to reduce the gap between a model's predicted confidence score and the actual observed predictive correctness, measuring the model's ability to estimate its predictive reliability. Whilst the confidence-based calibration of modern large-scale machine learning models has been actively investigated in recent years Guo et al. (2017); Minderer et al. (2021a); Achiam et al. (2023), calibration through the lens of prediction set size (PSS), aiming for a tight coupling between PSS and expected accuracy is under-investigated. Although the coverage of PSS is guaranteed, its alignment with prediction accuracy is also essential for making risk-aware decisions and makes conformal inference more reliable. As illustrated in Figure 1, point prediction calibration has already been thoroughly investigated, but it outputs a single prediction for each query thus cannot guarantee the coverage of ground truth in its prediction. Conformal prediction, a well-established approach for achieving

coverage guarantees, conveys uncertainty through the size of the prediction set Angelopoulos et al. (2021). Yet, it remains unclear whether this set size truly aligns with prediction correctness.

To fill in this gap, our work investigates how to build a model calibration framework for CP in multi-classification tasks. This calibration was mentioned in Lu et al. (2023) as an auxiliary experiment to visualize the correlation between the PSS and Top-1 accuracy qualitatively without a thorough study to build the connection between the PSS and accuracy. Meanwhile, CP calibration was studied systematically on regression tasks van der Laan & Alaa (2024), but our paper is the first attempt to systematically investigate the calibration of CP on classification

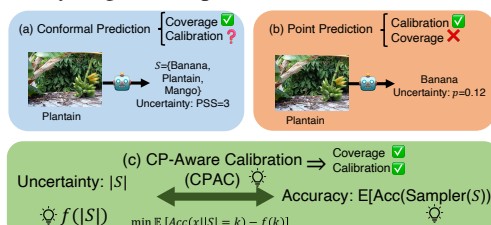

Figure 1: **(a)** and **(b)** compare CP and point prediction in coverage and calibration. **(c)** Two key contributions: *CP's calibration target in multinomial sampling* and *CP-aware calibration*.

tasks. Note that the fundamental difference between van der Laan & Alaa (2024) and our work is that van der Laan & Alaa (2024) treats point prediction and prediction interval independently, while we aim to calibrate a classification model so that the CP's prediction set is calibrated and has valid coverage at the same time. It is important to note that calibration for PSS is *fundamentally different* from conditional coverage Gibbs et al. (2025). While conditional coverage ensures subgroup-level validity, PSS calibration evaluates whether smaller sets consistently correspond to higher per-instance reliability, a property not captured by conditional guarantees. Also note that, studying whether the uncertainty CP conveys in practice (via set size) is trustworthy does not suggest CP should replace probabilistic calibration such as entropy or confidence.

Developing calibration for conformal prediction through PSS has three key challenges. First, while CP produces a set of plausible labels, it does not directly yield a point prediction and thus accuracy. Second, the function between PSS and prediction accuracy is not straightforward, compared to the linear function in traditional confidence-based uncertainty. Third, it is unclear how to effectively calibrate a model to ensure that smaller prediction sets consistently correspond to higher accuracy. To address these challenges, we first use multinomial sampling with temperature to generate a point prediction from a prediction set, enabling us to obtain the accuracy. Then, we introduce a calibration target function based on the predictive distribution that maps PSS to expected accuracy, capturing the relationship between uncertainty and reliability in CP. An empirical study on the calibration target functions reveals weak calibrations of conformalized models, highlighting the need for correction. To this end, we propose a CP-aware calibration algorithm based on bi-level optimization as a *pre-processing* step before the quantile computation in the CP framework. Our contributions are three-fold:

- We establish a connection between the PSS and accuracy by sampling a label from the prediction set with the predictive distribution. An empirical study on the alignment of PSS and accuracy demonstrates the weak calibration of PSS.

- We propose a calibration target function motivated by both our empirical study and a theoretical analysis of the predictive distribution. It can handle prediction sampling with different temperatures and has a lower calibration error on average compared with other alternative target functions.

- We propose a CP-aware calibration algorithm as a pre-processing step of CP to improve the calibration of CP. The effectiveness in classification tasks is validated using three benchmark datasets in computer vision and natural language understanding with state-of-the-art models, including vision transformers and GPT-2.

## 2 RELATED WORK

**Conformal Prediction and Uncertainty Quantification**. Different from existing methods in the frequentist's approach to prediction uncertainty, CP is distribution-free and can be applied to any black-box machine learning model as long as the data exchangeability assumption is satisfied Vovk et al. (1999); Shafer & Vovk (2008). The original version of CP needs to train a model multiple times but is later improved by Vovk et al. (2005) as the split conformal prediction that can be used in any black-box model, leading to its popularity in many applications Romano et al. (2020); Angelopoulos et al. (2021). Existing research mainly aims to improve the coverage validity Gibbs & Candes (2021) and efficiency Angelopoulos et al. (2021); Ghosh et al. (2023a;b); Liu et al. (2024), as well as extend

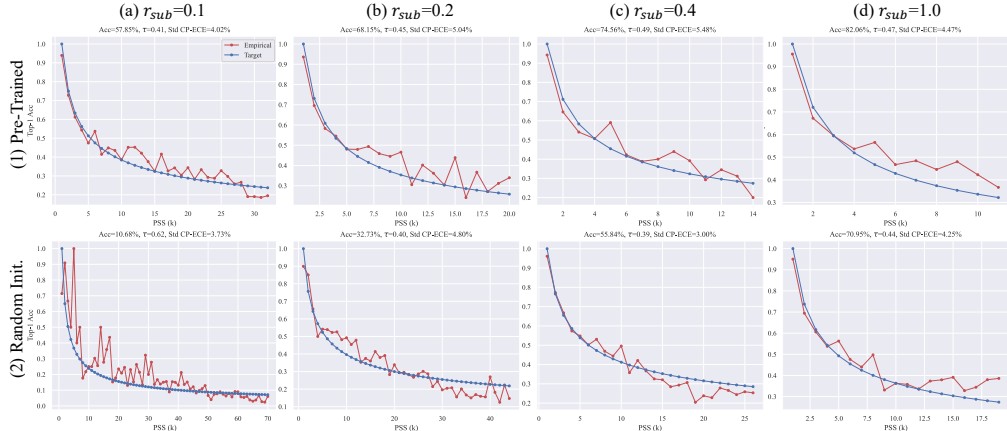

Figure 2: Reliability diagrams of ResNet50 trained on CIFAR100. The first row shows plots for varying training data sizes ($r_{sub}$ is the subsampling ratio) with a pre-trained (on ImageNet-1k) initial model, while the second row shows plots for a randomly initialized model. The **red** curve is the observed one while the **blue** curve is the target curve.

CP to non-exchangeable data Barber et al. (2023) and achieve conditional coverage Gibbs et al. (2023). More recently, a random set method with improved calibration is proposed Manchingal et al. (2025), but it cannot be readily plugged into a pre-trained model as it is developed for Bayesian neural networks. In contrast, our work systematically investigates the calibration of quantified uncertainty by CP in classification tasks. This topic has been studied by Lu et al. (2023) and van der Laan & Alaa (2024), but Lu et al. (2023) proposes federated CP for distributed users and does not solely focus on the calibration, and van der Laan & Alaa (2024) only considers regression tasks. The proposed conformalizing Venn-Abers calibration van der Laan & Alaa (2024) for regression models cannot be applied to our classification problem as they produce the calibration multi-prediction and prediction interval separately, but we aim to calibrate the prediction interval/set directly. The effect of temperature scaling in CP is also investigated Xi et al. (2024); Dabah & Tirer (2024), but they only consider the coverage and efficiency of PSS instead of reducing the gap between PSS and its accuracy

**Model Calibration**. Model calibration focuses on adjusting predictive models to ensure that the predictive uncertainty accurately reflects the true likelihood Vaicenavicius et al. (2019). Model calibration is crucial in safety-critic applications Huang et al. (2020) where decision-making relies on well-calibrated probabilities. Deep neural networks have been found to be weakly calibrated can be fixed by traditional methods like Platt scaling Guo et al. (2017), which fits a temperature scalar to a classifier's scores. Recent studies Minderer et al. (2021b) have shown that the large-scale pre-trained models are more calibrated, in particular for the convolution architecture. With the huge impact of large language models (LLMs), their calibration are also actively investigated Achiam et al. (2023); Xiong et al. (2024). However, the model calibration mainly focuses on the heuristic uncertainty such as confidence scores in classification. Our work aims to unveil the calibration of uncertainty when CP is used in state-of-art models including both vision transformer models for vision tasks and an LLM for language understanding tasks.

## 3 PRELIMINARIES

We introduce the necessary mathematical annotations and background in this section.

**Notations**. We split the dataset into three subsets, i.e., training set $\mathcal{D}_{tr} = \{(\boldsymbol{x}_i, y_i)\}_{i=1}^{N_{tr}}$, calibration set $\mathcal{D}_{cal} = \{(\boldsymbol{x}_i, y_i)\}_{i=1}^{N_{cal}}$ and test set $\mathcal{D}_{te} = \{(\boldsymbol{x}_i, y_i)\}_{i=1}^{N_{te}}$. A classification model is trained on $\mathcal{D}_{tr}$, and conformalization including Platt scaling is performed using the calibration set, and then the conformalized model is evaluated on the test set. Each data point $(\boldsymbol{x}, y)$ is sampled from a distribution over the data space $\mathcal{X} \times \mathcal{Y}$. As we only investigate the classification task in this paper, the label space $\mathcal{Y} = \{1, \cdots, K\}$ denoted as $[K]$ for simplicity. After training a deep classification model $f_\theta(\boldsymbol{x})$ with parameters $\theta$, the model produces a logit vector $\boldsymbol{l}_i \in \mathbb{R}^K$ for a test sample $\boldsymbol{x}_i$, where the $\arg\max_j \boldsymbol{l}_{ij}$ is the predicted label. The predictive distribution $\boldsymbol{p}_i$ is the output of the softmax function when the input is $\boldsymbol{l}_i$.

**Conformal Prediction**. Conformal prediction ensures population-level coverage guarantees without distributional assumptions and applies to both regression and classification. This study focuses on classification, where a conformalized classification model generates a prediction set $\mathcal{S}_i \in 2^{[K]}$ for a test sample $\boldsymbol{x}_i$ so that the coverage guarantee is ensured

$$P(y \in \mathcal{S}(\boldsymbol{x})) \geq 1 - \alpha, \tag{1}$$

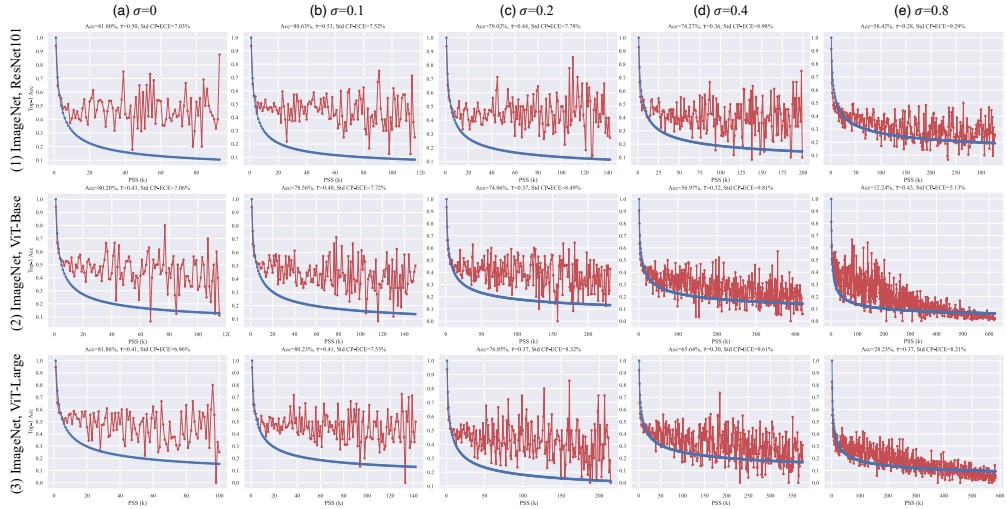

Figure 3: Reliability diagrams of three classification models on ImageNet when there is input noise sampled from Gaussian distribution with a standard deviation of $\sigma$.

where $1 - \alpha$ is a confidence level such as 90%, indicating that the prediction set will contain the ground-truth label with 90% confidence at the population level. The original CP needs to train a model multiple times to obtain such a guarantee, while our paper uses the *split conformal prediction* approach that does not need to train multiple times and can be plugged into any pre-trained black-box classifier Papadopoulos et al. (2002); Lei et al. (2018).

In this study, we use the APS (Adaptive Prediction Sets) method Romano et al. (2020) to perform the conformalization if not specified otherwise. There are two stages in APS, both done on $\mathcal{D}_{cal}$ (a) temperature scaling Guo et al. (2017) and (b) computing the $(1 - \alpha)$-quantile of conformity scores. The temperature scaling aims to make the confidence score more calibrated by find an optimal temperature to scale the logic vectors such that the likelihood of $\mathcal{D}_{cal}$ is maximized. After scaling the logits, conformity scores on $\mathcal{D}_{cal}$ are computed and their $(1 - \alpha)$-quantile can be found. We give a description on the process of computing the conformity scores and the quantile in Appendix B.

## 4    CALIBRATION OF CONFORMAL PREDICTION

We now introduce a definition of CP calibration and describe how to use multinomial sampling to obtain accuracy. This is followed by a theoretical analysis on the proposed calibration target function. Moreover, we propose a calibration algorithm for CP on classification tasks (Algorithm 1).

### 4.1    CALIBRATION OF A CONFORMALIZED MODEL

Calibration for the uncertainty expressed by confidence scores is straightforward, as prediction accuracy with a confidence score is easily obtained. However, as the uncertainty in CP is measured by the PSS, the connection between a prediction set and its prediction quality cannot be immediately obtained. In other words, a prediction set is not directly comparable to the ground-truth class. To build a connection between a prediction set and its prediction accuracy, we propose a multinomial sampling strategy to produce a prediction from the prediction set and take the average accuracy of the sampled labels as the prediction correctness. Denote the normalized predictive probability in the prediction set $\mathcal{S}_i$ as $\tilde{\boldsymbol{p}}_i = [\tilde{p}_{i1}, \cdots, \tilde{p}_{i|\mathcal{S}_i|}]$ where $\sum_j \tilde{p}_{ij} = 1$, and the multinomial distribution is a function of the predictive probability

$$\boldsymbol{q}_i^{(t)} = [\tilde{p}_{i1}^t, \cdots, \tilde{p}_{i|\mathcal{S}_i|}^t] / \sum_j^{|\mathcal{S}_i|} \tilde{p}_{ij}^t, \tag{2}$$

where $t \in [0, +\infty)$ is the exponent for the sampling. When $t = 0$, we use uniform sampling to produce the predictive label. When $t$ approaches $+\infty$, the sampling is equivalent to Top-1 accuracy using the maximum confidence. With the sampled accuracy, we give a definition for the CP calibration.

**Definition 4.1.** A classifier is conformally calibrated if the conditional expectation of accuracy using multinomial sampling with the temperature $t$ decreases with the prediction set size $k$, i.e.,

$$\mathbb{E}[Acc_t(\boldsymbol{x})|S(\boldsymbol{x}) = k] = f(k), \tag{3}$$

where $f(k)$ is a monotonically decreasing function and $S(\cdot)$ maps an input sample into its PSS, the condition means for the expectation is computed on all $\boldsymbol{x}$ with $S(\boldsymbol{x}) = k$, and

$$Acc_t(\boldsymbol{x}) = \mathbb{E}_{\boldsymbol{q}^{(t)}(\boldsymbol{x})} \mathbf{1}(\hat{y} = y). \tag{4}$$

We define the following two metrics for the calibration error for a conformalized model.

$$\text{Standard CP-ECE} = \sum_{k=1}^{K} \frac{N_{te}^{(k)}}{N_{te}} |Acc_t(\boldsymbol{x}|S(\boldsymbol{x}) = k) - f(k)|, \quad \text{Uniform CP-ECE} = \sum_{k=1}^{K} \frac{1}{K} |Acc_t(\boldsymbol{x}|S(\boldsymbol{x}) = k) - f(k)|. \quad (5)$$

The standard CP Expected Calibration Error (CP-ECE) is weighted by the proportion of samples with PSS equal to $k$ relative to the entire test set. In contrast, the uniform CP-ECE corresponds to the curve-fitting error measured by the absolute distance. We use the uniform CP-ECE since we want to measure the curve fitting performance in the reliability diagram without considering the number of samples in each bin. Moreover, we believe it is an important metric in practice as well, because it gives the same weight to different groups to prevent discrimination towards minor groups Mehrabi et al. (2021). It is also similar to the unweighted accuracy that is often used to measure the performance of a model as a complement to the standard weighted accuracy. While both our work and Huang et al. (2024) define calibration, note that they are fundamentally different, as we focus on PSS calibration within conformal prediction and propose an optimization algorithm to reduce calibration error, whereas Huang et al. (2024) neither considers conformal prediction nor minimizes calibration error.

## 4.2 CALIBRATION TARGET FUNCTION

Another challenge for CP calibration is the target calibration curve $f(k)$. In confidence calibration, the identity $f(c) = c$ is the target curve as the perfect calibration Guo et al. (2017) is defined by

$$\mathbb{P}(\hat{Y} = Y | \hat{P} = p) = p, \forall p \in [0, 1]. \quad (6)$$

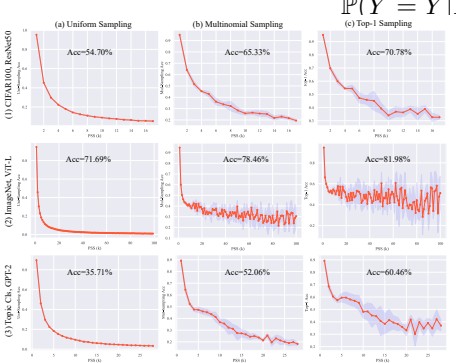

Figure 4: Sampling accuracy versus prediction set size (PSS) on three datasets. As the temperature in multinomial sampling decreases (uniform⇒multinomial⇒Top-1), accuracy increases but the calibration of PSS worsens. The shaded area shows the standard deviation over five seeds.

However, the PSS from CP does not have a straightforward relationship with the prediction correctness except for the monotonically decreasing property. We propose a calibration target in this work, motivated by the heuristic generalization of the uniform sampling curve but also derived from a theoretical analysis based on a Dirichlet distribution assumption on the predictive probability.

We start from a simple case, where we use a uniform weight to sample from the predictive probability, i.e., the temperature in multinomial sampling is zero. We assume that the re-normalized predictive probability $\tilde{p}_{ij}$ in the prediction set $\mathcal{S}_i$ is the probability of $j$th class being the true class, then the expected accuracy of using multinomial sampling with weight $\boldsymbol{q}_i$ is

$$Acc_t(\boldsymbol{x}_i) = \sum_{j \in \mathcal{S}_i} q_{ij}^{(t)} \tilde{p}_{ij}, \quad (7)$$

where $\sum_{j=1}^{|\mathcal{S}_i|} q_{ij}^{(t)} = 1, \sum_{j=1}^{|\mathcal{S}_i|} \tilde{p}_{ij} = 1$. Note that the binary correctness in Equ. 4 becomes a probabilistic one. It is straightforward to obtain that if $q_{ij} = 1/|\mathcal{S}_i|$ for every $j$, then the expected accuracy is $1/|\mathcal{S}_i|$. Fig. 4 shows that the curve of accuracy versus PSS fits well with the power function $f(k) = 1/k$, validating the assumption of the true class distribution.

**General cases.** Inspired by the success of the power function in the uniformly weighted sampling, we propose to use a power function $f(k) = 1/k^\tau$ as the calibration target where $\tau \in [0, 1]$, to accommodate the improvement in accuracy when using non-uniformly weighted sampling. Intuitively, when we use a low temperature in the multinomial sampling, the probability distribution gets sparse and the effective set size decreases. To account for such a set size decrease in the calibration function, we use $\tau < 1$ to decrease it from $|\mathcal{S}_i|$ to $|\mathcal{S}_i|^\tau$. Two alternative functions are the exponential decay $\exp(-\tau(k - 1))$ and the logarithmic scaling function $1/(1 + \tau \log(k))$, where $\tau > 0$. However, our empirical study in Sec. 5 shows that the exponential decay function has a much faster decreasing rate than the power function, while the logarithmic scaling function cannot fit the curve of uniform sampling well.

We give a theorem on the relationship between the expected accuracy and the target calibration function by assuming that both $\boldsymbol{p}$ and $\boldsymbol{q}$ are sampled from two Dirichlet distributions with the same underlying shape.

**Theorem 4.2** (Expected Accuracy and Prediction Set Size). *Let $K \geq 2$ be a dimension, and let $\boldsymbol{a} = (a_1, \ldots, a_K)$ be a vector with $a_j \geq 0$ and $\sum_{j=1}^{K} a_j = 1$. Suppose $\alpha_0 > 0$ and $\beta_0 > 0$ are*

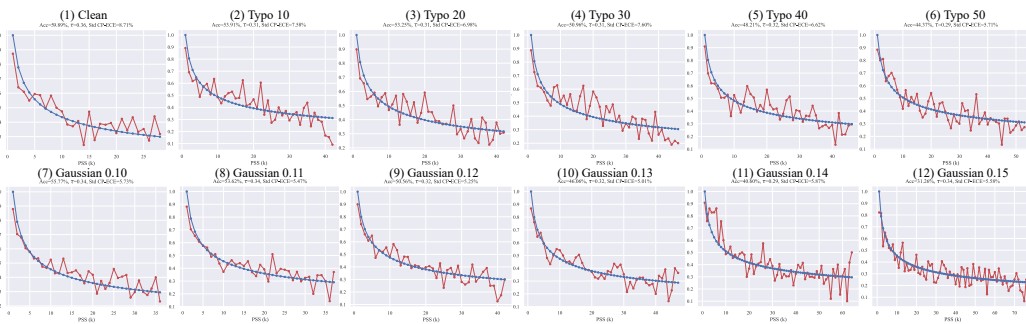

Figure 5: Reliability diagrams of GPT-2 on a topic classification dataset when there is Gaussian noise added on the embeddings or typos in textual input.

---

**Algorithm 1** CP-Aware Calibration

---

**Require:** Calibration dataset $\mathcal{D}_{\text{cal}}$, logits from a pre-trained deep neural network $\{l_i\}_i^{N_{cal}}$, regularization parameter $\lambda$, target calibration parameter $\tau$, learning rate $\eta$, sampling temperature $t > 0$, batch size $B$, optimization round $M_{opt}$, mis-coverage level $\alpha$

1: **for** $e \leftarrow 1 : M_{opt}$ **do**
2:   Run $g(\boldsymbol{W}, \boldsymbol{b}) = \nu$, i.e., Compute conformality scores $\{E_i\}_{i=1}^{N_{cal}}$ and find the $(1 - \alpha)$-quantile $\nu$
3:   Randomly divide $\mathcal{D}_{cal}$ into $T$ batches $\{\mathcal{B}_{it}\}_{it=1}^T$ with batch size $B$
4:   **for** $it \leftarrow 1 : T$ **do**
5:     Compute $\nabla_{\boldsymbol{W},\boldsymbol{b}} \mathcal{L}_{ps} + \lambda(\|\boldsymbol{W} - \boldsymbol{I}\|_2^2 + \|\boldsymbol{b}\|_2^2)$ for $i \in \mathcal{B}_t$, denoted as $(\Delta\boldsymbol{W}, \Delta\boldsymbol{b})$
6:     $(\boldsymbol{W}_{it}, \boldsymbol{b}_{it}) \leftarrow (\boldsymbol{W}_{it-1}, \boldsymbol{b}_{it-1}) - \eta\Delta(\boldsymbol{W}, \boldsymbol{b})$
7:   **end for**
8: **end for**
9: **return** The calibration parameter $(\boldsymbol{W}, \boldsymbol{b})$

---

*positive scalars, and define Dirichlet parameters*
$$\alpha = (\alpha_1, \dots, \alpha_K), \quad \beta = (\beta_1, \dots, \beta_K)$$
*where $\alpha_j = \alpha_0 \, a_j$ and $\beta_j = \beta_0 \, a_j$. Let $\boldsymbol{p} \sim \text{Dir}(\alpha)$ and $\boldsymbol{q} \sim \text{Dir}(\beta)$ be independent draws. We have*
$$\mathbb{E}[\boldsymbol{q} \cdot \boldsymbol{p}] = \frac{1}{K^\tau}, \quad where \quad \tau = -\frac{\log(\sum_{j=1}^K a_j^2)}{\log K}.$$

**Remark.** This theorem assumes that $\boldsymbol{p}$ and $\boldsymbol{q}$ are drawn from two Dirichlet distributions with the same underlying mass distribution vector $\boldsymbol{a}$. This assumption is due to the proposed multinomial sampling, where both sampling accuracy $\boldsymbol{q}^{(t)}$ and correctness probability $\tilde{\boldsymbol{p}}$ follow the same mass distribution. The difference between $\text{Dir}(\alpha)$ and $\text{Dir}(\beta)$ is the degree of concentration, which corresponds to multinomial sampling with temperature. The exponent $\tau$ is determined by the shape of the vector $\boldsymbol{a}$. When $\boldsymbol{a}$ is close to uniform, the expected accuracy is $1/K$. When $\boldsymbol{a}$ is highly peaked at a single dimension, the expected accuracy will be one. We report the calibration error of more models on three datasets in Sec. 5 using this target calibration function. Note that we derived a similar decay function when the Dirichlet distribution does not hold and the prediction is sampled from a logistic-normal distribution with ordered variances, see Appendix A. Thus, both the Dirichlet and logistic-normal distributions serve as illustrative instantiations of our target behavior rather than restrictive assumptions.

### 4.3 CONFORMAL-PREDICTION-CALIBRATION WITH BI-LEVEL OPTIMIZATION

As the calibration error of CP is not satisfactory, particularly on challenging datasets such as ImageNet as our experiments show, we propose to post-process the model prediction so that the uncertainty from CP is better aligned with the accuracy.

Denote the logit vector of $i$th sample in the calibration set as $\boldsymbol{l}_i$, the model is calibrated by optimizing the weight matrix $\boldsymbol{W}$ and bias $\boldsymbol{b}$ similar to Platt scaling, but with our proposed calibration target. The correctness probability $\tilde{\boldsymbol{p}}$ and sampling probability $\boldsymbol{q}^{(t)}$ function is

$$\tilde{l}_{ij} = \begin{cases} [\boldsymbol{W}^T \boldsymbol{l}_i + \boldsymbol{b}]_j, \text{if } j \in \mathcal{S}_i \\ -\infty, \text{if } j \notin \mathcal{S}_i \end{cases}, \tilde{\boldsymbol{p}}_i = \text{softmax}(\tilde{l}_i), \quad \boldsymbol{q}_i = [\tilde{p}_{i1}^t, \cdots, \tilde{p}_{ik}^t] / \sum_j^{|\mathcal{S}_i|} \tilde{p}_{ij}^t \quad (8)$$

| | | Std CP-ECE | Uni CP-ECE | Acc. | Cov. | PSS | | | Std CP-ECE | Uni CP-ECE | Acc. | Cov. | PSS |
|---|---|---|---|---|---|---|---|---|---|---|---|---|---|
| Clean | PS | 6.88(0.06) | 11.02(0.30) | 81.98(0.08) | 94.52(0.13) | 18.47(0.64) | Clean | PS | 7.16(0.10) | 10.01(0.27) | 80.35(0.09) | 94.18(0.17) | 18.05(0.90) |
| | PS-Full | 6.38(0.06) | 8.40(0.50) | 81.98(0.08) | 93.88(0.07) | 13.70(0.25) | | PS-Full | 6.65(0.07) | 9.36(0.44) | 80.35(0.09) | 93.75(0.14) | 14.44(0.54) |
| | CPAC | 6.74(0.09) | 6.74(0.47) | 80.17(0.16) | 92.39(0.08) | 10.04(0.10) | | CPAC | 7.45(0.11) | 7.09(0.54) | 77.68(0.14) | 91.29(0.09) | 10.43(0.53) |
| Norm-0.1 | PS | 7.52(0.05) | 10.70(0.28) | 80.39(0.11) | 94.22(0.14) | 20.71(0.77) | Norm-0.1 | PS | 7.78(0.09) | 9.16(0.30) | 78.67(0.06) | 93.98(0.22) | 20.78(1.19) |
| | PS-Full | 6.90(0.04) | 9.10(0.18) | 80.39(0.11) | 93.55(0.11) | 15.00(0.42) | | PS-Full | 7.21(0.10) | 9.12(0.47) | 78.67(0.06) | 93.43(0.18) | 16.19(0.58) |
| | CPAC | 7.21(0.14) | 7.39(1.12) | 78.73(0.16) | 91.83(0.19) | 10.62(0.33) | | CPAC | 7.72(0.11) | 7.17(0.58) | 75.45(0.32) | 90.75(0.11) | 11.37(0.26) |
| Norm-0.2 | PS | 8.30(0.11) | 10.37(0.21) | 76.98(0.07) | 93.85(0.18) | 27.49(1.15) | Norm-0.2 | PS | 8.53(0.10) | 9.98(0.35) | 74.13(0.05) | 93.57(0.12) | 31.27(0.76) |
| | PS-Full | 7.67(0.13) | 10.54(0.48) | 76.98(0.07) | 93.04(0.17) | 20.03(0.55) | | PS-Full | 7.97(0.07) | 10.11(0.29) | 74.13(0.05) | 92.93(0.06) | 24.89(0.29) |
| | CPAC | 7.99(0.12) | 7.99(0.84) | 75.33(0.11) | 91.31(0.19) | 14.90(0.45) | | CPAC | 8.46(0.16) | 7.98(0.36) | 70.57(0.41) | 89.66(0.31) | 17.11(0.40) |
| Norm-0.4 | PS | 9.73(0.16) | 8.82(0.17) | 65.69(0.03) | 92.85(0.20) | 58.44(1.48) | Norm-0.4 | PS | 9.75(0.14) | 7.21(0.16) | 57.19(0.16) | 92.05(0.07) | 82.51(0.40) |
| | PS-Full | 9.34(0.07) | 8.69(0.28) | 65.69(0.03) | 91.93(0.14) | 48.13(1.01) | | PS-Full | 9.59(0.10) | 7.54(0.17) | 57.19(0.16) | 91.29(0.10) | 75.44(0.78) |
| | CPAC | 9.02(0.11) | 8.30(0.64) | 64.00(0.15) | 89.72(0.17) | 38.30(1.28) | | CPAC | 8.67(0.10) | 7.24(0.36) | 53.43(0.28) | 87.77(0.41) | 63.31(3.75) |
| Norm-0.8 | PS | 8.09(0.10) | 5.30(0.08) | 28.34(0.09) | 90.60(0.21) | 239.18(2.99) | Norm-0.8 | PS | 5.06(0.12) | 7.25(0.27) | 12.30(0.06) | 90.00(0.30) | 429.42(5.39) |
| | PS-Full | 8.29(0.20) | 5.43(0.05) | 28.34(0.09) | 90.37(0.27) | 258.33(4.55) | | PS-Full | 5.32(0.07) | 5.10(0.16) | 12.30(0.06) | 89.84(0.28) | 469.92(5.28) |
| | CPAC | 7.26(0.13) | 4.94(0.08) | 28.17(0.19) | 88.95(0.16) | 239.63(2.55) | | CPAC | 4.39(0.23) | 4.72(0.27) | 12.50(0.21) | 89.09(0.19) | 437.94(3.66) |
| Blur-3 | PS | 7.88(0.10) | 11.04(0.56) | 79.03(0.07) | 93.92(0.14) | 23.30(0.89) | Blur-3 | PS | 8.17(0.10) | 10.13(0.46) | 77.06(0.10) | 93.61(0.19) | 24.61(0.90) |
| | PS-Full | 7.50(0.05) | 11.50(0.49) | 79.03(0.07) | 93.52(0.10) | 19.37(0.51) | | PS-Full | 7.79(0.09) | 10.68(0.41) | 77.06(0.10) | 93.27(0.16) | 21.24(0.78) |
| | CPAC | 7.57(0.15) | 8.38(0.56) | 77.62(0.15) | 92.15(0.05) | 14.67(0.30) | | CPAC | 8.02(0.07) | 7.79(0.39) | 75.19(0.32) | 91.25(0.10) | 15.18(0.47) |
| Blur-5 | PS | 8.40(0.07) | 10.64(0.28) | 77.94(0.05) | 93.74(0.18) | 25.92(1.19) | Blur-5 | PS | 8.76(0.05) | 10.20(0.44) | 75.24(0.09) | 93.50(0.14) | 28.73(0.70) |
| | PS-Full | 8.01(0.09) | 10.61(0.50) | 77.94(0.05) | 93.39(0.11) | 22.50(0.58) | | PS-Full | 8.42(0.10) | 10.08(0.34) | 75.24(0.09) | 93.21(0.17) | 25.44(0.84) |
| | CPAC | 8.02(0.19) | 8.51(0.22) | 76.41(0.20) | 92.05(0.09) | 17.24(0.57) | | CPAC | 8.39(0.16) | 8.07(0.15) | 73.09(0.10) | 91.01(0.21) | 18.73(0.50) |
| Blur-7 | PS | 8.59(0.06) | 10.27(0.21) | 77.51(0.05) | 93.67(0.23) | 27.08(1.22) | Blur-7 | PS | 8.96(0.16) | 9.74(0.27) | 74.39(0.06) | 93.40(0.16) | 31.02(0.88) |
| | PS-Full | 8.24(0.08) | 10.93(0.31) | 77.51(0.05) | 93.37(0.13) | 23.83(0.51) | | PS-Full | 8.67(0.12) | 9.91(0.51) | 74.39(0.06) | 93.09(0.19) | 27.65(0.97) |
| | CPAC | 8.12(0.18) | 8.96(0.44) | 76.25(0.15) | 92.10(0.07) | 18.44(0.59) | | CPAC | 8.54(0.14) | 8.11(0.51) | 72.12(0.16) | 90.82(0.21) | 20.55(0.56) |
| Drop-1 | PS | 9.69(0.06) | 10.99(0.46) | 74.50(0.13) | 93.55(0.19) | 40.62(2.10) | Drop-1 | PS | 9.76(0.15) | 9.46(0.47) | 71.04(0.07) | 93.10(0.18) | 39.13(1.03) |
| | PS-Full | 9.00(0.10) | 11.38(0.45) | 74.50(0.13) | 92.54(0.13) | 29.33(0.82) | | PS-Full | 9.33(0.14) | 9.87(0.50) | 71.04(0.07) | 92.41(0.24) | 31.90(0.94) |
| | CPAC | 8.71(0.22) | 9.52(0.32) | 73.68(0.19) | 91.32(0.18) | 23.93(0.67) | | CPAC | 9.04(0.12) | 8.22(0.69) | 68.26(0.14) | 89.72(0.18) | 25.09(0.33) |
| Drop-3 | PS | 11.05(0.13) | 7.70(0.25) | 60.83(0.11) | 92.46(0.12) | 93.09(1.16) | Drop-3 | PS | 10.31(0.15) | 6.60(0.09) | 54.91(0.11) | 91.93(0.24) | 93.11(2.37) |
| | PS-Full | 11.37(0.12) | 8.71(0.31) | 60.83(0.11) | 91.15(0.16) | 79.22(1.73) | | PS-Full | 10.62(0.16) | 7.29(0.17) | 54.91(0.11) | 91.00(0.32) | 86.53(2.79) |
| | CPAC | 10.09(0.21) | 8.28(0.24) | 61.99(0.09) | 91.82(0.19) | 70.12(1.43) | | CPAC | 8.09(0.46) | 8.33(0.82) | 50.49(1.14) | 88.08(0.20) | 84.83(10.46) |
| Drop-5 | PS | 10.29(0.03) | 6.30(0.15) | 44.36(0.12) | 91.77(0.12) | 161.52(1.21) | Drop-5 | PS | 8.75(0.08) | 5.42(0.20) | 35.38(0.07) | 91.03(0.23) | 187.77(3.32) |
| | PS-Full | 10.77(0.10) | 6.61(0.08) | 44.36(0.12) | 90.69(0.08) | 169.74(1.20) | | PS-Full | 9.18(0.05) | 5.63(0.05) | 35.38(0.07) | 90.48(0.28) | 211.86(3.96) |
| | CPAC | 9.67(0.09) | 6.18(0.21) | 45.01(0.20) | 88.98(0.13) | 126.75(1.51) | | CPAC | 6.81(0.54) | 5.35(0.09) | 32.77(0.53) | 88.62(0.36) | 215.71(6.65) |
| Drop-7 | PS | 6.73(0.11) | 5.62(0.20) | 19.69(0.05) | 90.48(0.32) | 306.40(4.44) | Drop-7 | PS | 5.10(0.07) | 7.16(0.11) | 11.97(0.09) | 89.97(0.12) | 423.58(2.71) |
| | PS-Full | 7.71(0.15) | 5.21(0.09) | 19.69(0.05) | 90.18(0.27) | 392.71(6.12) | | PS-Full | 5.84(0.10) | 5.26(0.16) | 11.97(0.09) | 89.90(0.26) | 490.89(5.01) |
| | CPAC | 6.93(0.11) | 4.86(0.16) | 21.77(0.10) | 88.84(0.40) | 300.52(8.65) | | CPAC | 4.78(0.17) | 4.84(0.21) | 12.73(0.22) | 88.95(0.14) | 427.64(2.94) |

Table 1: Result of ViT-Large (left) and ViT-Base (right) on ImageNet-1k. *Norm-$\sigma$* means Gaussian noise with a std $\sigma$, *Blur-n* means Gaussian blur with kernel size $n$ and *Drop-r* means randomly drop pixels with ratio $r$.

The optimization problem is

$$\min_{\boldsymbol{W},\boldsymbol{b}} \sum_i (\sum_{j \in \mathcal{S}_i} \tilde{p}_{ij}(\boldsymbol{W},\boldsymbol{b},\nu) q_{ij}^{(t)}(\boldsymbol{W},\boldsymbol{b},\nu) - f_\tau(|\mathcal{S}_i|))^2, \quad s.t. \quad \nu - g(\boldsymbol{W},\boldsymbol{b}) = 0, \qquad (9)$$

where the $f_\tau(\cdot)$ function is the target calibration curve with $\tau$ as the exponent. To optimize $(\boldsymbol{W},\boldsymbol{b})$, we need first to obtain the prediction set $\mathcal{S}_i$ for each sample by finding the empirical $(1-\alpha)$-quantile of conformity scores in the calibration set. We denote the target as $\nu$ and the searching function as $g$. Thus, we formulate the CP calibration problem as a bi-level optimization problem, where the prediction set is produced from solving the lower-level conformity scores' $(1-\alpha)$-quantile. As this objective function will lead to a zero gradient for samples of PSS=1, we use the cross-entropy loss for samples with PPS=1.

To solve this bi-level optimization problem, we adopt the alternative optimization approach as shown in Alg. 1 by assuming the $\nu$ variable does not change drastically during optimization $(\boldsymbol{W},\boldsymbol{b})$. Note that we add a regularization term to constrain the distance between $(\boldsymbol{W},\boldsymbol{b})$ and the initialization $(\boldsymbol{I},\boldsymbol{0})$ to prevent overfitting on the calibration set. Note that we choose to optimize the full weight matrix instead of a temperature parameter as in Guo et al. (2017) as our pilot empirical study shows that a single scalar does not affect the calibration significantly as it fails to re-rank the class probabilities.

## 5 EXPERIMENTAL RESULTS

This section first describes the experimental details and then reports our empirical study on the calibration of CP on the three datasets.

### 5.1 EXPERIMENTAL SETTINGS

We conducted the experiment on three datasets using seven models, including two for image classification and one for topic classification. The experimental details are reported below.

**CIFAR100 Krizhevsky et al. (2009).** The dataset comprises 100 categories, each containing 600 images where 500 of them are used for training and 100 are used for test. We use 20% of the original test data as the calibration set and the rest 80% as the test set. The model we use is ResNet50 He et al. (2016), pre-trained on ImageNet Deng et al. (2009) or randomly initialized He et al. (2015). We train the model for 60 epochs and decay the learning rate by dividing it by 10 at 30th and 50th epoch. The initial learning rate is 0.1 in all the CIFAR100 training trials.

**ImageNet-1k Deng et al. (2009).** The dataset consists of approximately 1.28 million training images and 50,000 validation images, categorized into 1,000 classes. We utilize three models—ResNet101 He et al. (2016), ViT-B, and ViT-L Dosovitskiy et al. (2021)—with parameter sizes of 44.5M, 86M, and 307M, respectively. All images are resized to $224 \times 224$, and the patch sizes in ViT-B and ViT-L are $16 \times 16$. These pre-trained models, officially released and trained on ImageNet-1K, are used without further modifications. We evaluate the models under three types of image perturbations: Gaussian noise, Gaussian blur, and pixel dropout. (**1**) Gaussian Noise: We apply Gaussian noise with four different sigma values ($\sigma$, the square root of variance): 0.1, 0.2, 0.4, and 0.8. Each sigma value is uniformly applied across all test images. Additionally, we test a range of sigma values (0

to 0.8), where a random sigma is sampled to each image individually. (**2**) Gaussian Blur: Images are perturbed using Gaussian blur with kernel sizes of 3×3, 5×5, and 7×7. (**3**) Pixel Dropout: We randomly drop pixels from images at varying ratios: 0.1, 0.3, 0.5, and 0.7.

**Topic Classification**[1]. The dataset includes 22,500 pieces of text which are categorized into 120 topics. We split the dataset with 80% as the training set and 20% as the test set to fine-tune a GPT-2 Small model (137M) Radford et al. (2019) for the topic classification task. In evaluation, we test the calibration performance of different methods on the clean and two perturbed datasets. The perturbation strategies are: (**1**) Gaussian Noise: We add a norm distribution noise $\epsilon \sim N(\mu, \sigma^2)$ on the text embeddings. We set $\mu$ to 0, and vary the perturbation strength by assigning $\sigma$ values in

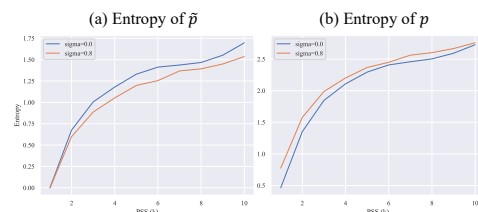

Figure 6: Different behaviour of predictive distributions within the prediction set and the whole dimension when using ImageNet and ResNet101.

the range [0.10, 0.15], incremented by 0.01; and (**2**) Typo: We randomly insert English keystroke typo, which is a mixture of insertion, deletion, substitution, and transposition Kukich (1992), to the test data to simulate the practical scenario. The perturbation rate, i.e., the portion of typo words relative to the total words, ranges from 10% to 50%, stepping by 10%.

**CPAC details.** We use 20% of the original test set as the calibration set $\mathcal{D}_{cal}$ in our experiment. Based on our preliminary experiment, the sampling temperature $t$ is 3, the round of optimization $M_{opt}$ is 4, the regularization hyperparameter $\lambda = 1e{-}4$ and the batch size is 1024. We use grid search to choose the optimal learning rate from $\{1e{-}4, 3e{-}4, 1e{-}3, 3e{-}3, 1e{-}2, 3e{-}2, 1e{-}1, 3e{-}1\}$ and $\tau$ from $\{0.1, 0.2, \cdots, 0.6\}$. The CPAC is performed on samples with low PSS (PSS<400 on ImageNet and PSS<70 on Topic Cls. data) as we only need to cover $(1-\alpha)$ of all samples in CP and we choose to optimize those low-PSS samples. We run the experiment five times in each setting by using five random seeds when splitting the original test set and report the average of the CP-ECE, accuracy, coverage and PSS. We use *CPAC* to denote our method and *PS* to denote the standard confidence calibration method in APS. All experiments were run on NVIDIA GeForce RTX 3090.

|  |  | Std CP-ECE | Uni CP-ECE | Acc. | Cov. | PSS |
|---|---|---|---|---|---|---|
| Clean | PS | 7.87(0.58) | 6.01(0.53) | 60.46(0.36) | 91.08(1.00) | 9.07(0.77) |
|  | PS-Full | 7.33(0.45) | 6.21(0.37) | 60.46(0.36) | 91.58(0.98) | 9.06(0.63) |
|  | CPAC | 7.02(0.55) | 5.50(0.46) | 60.40(0.38) | 91.71(0.81) | 9.18(0.59) |
| Norm-0.10 | PS | 6.36(0.81) | 5.29(0.75) | 56.13(0.28) | 90.64(1.25) | 12.10(1.47) |
|  | PS-Full | 6.48(0.56) | 5.72(0.71) | 56.13(0.28) | 90.84(1.26) | 11.88(1.37) |
|  | CPAC | 5.91(0.50) | 5.08(0.63) | 56.07(0.27) | 91.14(0.93) | 11.93(1.05) |
| Norm-0.11 | PS | 5.69(0.21) | 4.96(0.42) | 53.88(0.21) | 90.66(0.57) | 13.62(0.70) |
|  | PS-Full | 5.24(0.38) | 4.59(0.43) | 53.88(0.21) | 90.90(0.63) | 13.22(0.65) |
|  | CPAC | 5.32(0.43) | 4.59(0.49) | 53.88(0.25) | 90.35(0.92) | 12.62(0.94) |
| Norm-0.12 | PS | 5.28(0.05) | 5.22(0.32) | 50.90(0.29) | 90.89(0.29) | 17.06(0.49) |
|  | PS-Full | 5.06(0.51) | 5.15(0.59) | 50.90(0.29) | 91.03(0.52) | 16.47(1.00) |
|  | CPAC | 5.02(0.56) | 4.87(0.68) | 50.96(0.29) | 91.07(0.92) | 16.61(1.77) |
| Norm-0.13 | PS | 5.51(0.42) | 5.42(0.33) | 46.02(0.24) | 89.72(0.17) | 20.11(0.32) |
|  | PS-Full | 5.37(0.31) | 5.43(0.22) | 46.02(0.24) | 89.77(0.33) | 19.64(0.69) |
|  | CPAC | 5.24(0.18) | 5.19(0.30) | 46.05(0.24) | 89.49(0.35) | 18.98(0.54) |
| Norm-0.14 | PS | 5.31(0.43) | 6.04(0.57) | 39.46(0.70) | 90.13(1.43) | 30.86(3.72) |
|  | PS-Full | 4.95(0.42) | 5.56(0.56) | 39.46(0.70) | 90.10(1.47) | 30.45(3.77) |
|  | CPAC | 5.10(0.36) | 5.87(0.62) | 39.37(0.78) | 89.86(1.79) | 30.01(4.72) |
| Norm-0.15 | PS | 5.49(0.14) | 6.18(0.32) | 31.06(0.28) | 89.85(0.67) | 40.83(1.73) |
|  | PS-Full | 5.21(0.56) | 6.01(0.62) | 31.06(0.28) | 89.89(0.64) | 40.99(1.83) |
|  | CPAC | 4.69(0.32) | 5.46(0.22) | 31.10(0.35) | 89.99(0.68) | 41.12(2.01) |
| Typo-10 | PS | 7.69(0.46) | 6.74(0.20) | 54.35(0.27) | 91.02(0.58) | 15.17(0.75) |
|  | PS-Full | 7.01(0.37) | 6.07(0.17) | 54.35(0.27) | 90.92(0.59) | 14.42(0.71) |
|  | CPAC | 6.48(0.40) | 5.40(0.24) | 54.66(0.47) | 91.27(0.44) | 14.78(0.64) |
| Typo-20 | PS | 7.79(0.53) | 6.36(0.34) | 53.09(0.17) | 90.37(0.53) | 15.38(0.65) |
|  | PS-Full | 6.84(0.34) | 5.87(0.26) | 53.09(0.17) | 90.42(0.57) | 14.73(0.61) |
|  | CPAC | 6.48(0.48) | 5.73(0.50) | 53.41(0.44) | 90.99(0.21) | 15.65(0.31) |
| Typo-30 | PS | 7.24(0.48) | 6.63(0.67) | 51.54(0.40) | 90.35(0.51) | 17.62(0.62) |
|  | PS-Full | 6.10(0.62) | 5.75(0.85) | 51.54(0.40) | 90.66(0.52) | 17.12(0.80) |
|  | CPAC | 6.48(0.35) | 6.35(0.48) | 51.61(0.39) | 90.95(0.33) | 17.33(0.76) |
| Typo-40 | PS | 6.36(0.53) | 5.94(0.49) | 48.14(0.31) | 90.59(0.45) | 19.66(0.79) |
|  | PS-Full | 5.83(0.39) | 5.67(0.34) | 48.14(0.31) | 90.75(0.49) | 19.33(1.09) |
|  | CPAC | 5.97(0.54) | 5.67(0.32) | 48.10(0.47) | 90.16(0.60) | 18.25(1.00) |
| Typo-50 | PS | 5.34(0.41) | 5.28(0.53) | 44.44(0.26) | 89.85(0.80) | 23.55(1.85) |
|  | PS-Full | 5.08(0.63) | 5.13(0.76) | 44.44(0.26) | 89.96(0.65) | 23.31(1.60) |
|  | CPAC | 5.00(0.30) | 5.14(0.39) | 44.47(0.31) | 90.17(0.41) | 23.48(0.70) |

Table 2: Result of GPT-2 on Topic Classification. Norm means adding Normal distribution noise on embedding vectors and Typo means mixing typos with original text.

As high-temperature sampling tends to have a good calibration error, we mainly investigate the most ill-behaved sampling strategy, i.e., Top-1 sampling, in our experiment. All figures and tables show the result of Top-1 accuracy, if not specified otherwise. During the test stage, we use grid search to find the optimal $\tau$ to compute the standard and uniform CP-ECE respectively. In reliability diagrams (Accuracy versus PSS), we only visualize the result of one random seed following the convention in Guo et al. (2017). Note that either sampling or expectation is possible to report but we use the sampling notion to approximate the real-world decision-making process in this paper. We exclude the empty PSS case in our implementation by setting the minimum PSS to be 1.

### 5.2 TARGET CALIBRATION FUNCTION

We compare the curve fitting error of using the proposed target function and two alternatives, exponential function $\exp(-\tau(k-1))$ and logarithmic function $1/(1 + \tau \log(k))$ in Tab. 3. The logarithmic function is better than the power function in Multinomial and Top-1 sampling, but it fails to fit the simple curve of uniform sampling. Therefore, we still use the power function in our paper but the logarithmic function can also be used in low-temperature sampling.

[1] https://huggingface.co/datasets/valurank/Topic_Classification

### 5.3 Factors that Affect CP Calibration

**Pre-Trained Model vs. Random Initialization.** Fig. 2.1 and 2.2 compare the reliability diagram of with and without pre-trained weights on CIFAR100. In all cases, the standard CP-ECE of using pre-trained weights is worse than using random initialization, despite its improved accuracy. In terms of the target function, when there is sufficient data, i.e., subsampling ratio is 0.2, 0.4 or 0.8, $\tau$ of pre-trained weights is larger than random initialization, meaning that the predictive distribution of using pre-trained weights is more uniform than that of random initialization. However, when there is only limited data, i.e., subsampling ratio is 0.1, $\tau$ is less peaked in random initialization than in pre-trained weights with very low accuracy. This corroborates the effectiveness of a pre-trained model in the low-data regime, but also shows its weakness in CP calibration.

**Subsampling.** Fig. 2.a-d show the change of reliability diagrams when more training data is used. An increasing trend in accuracy from left to right is observed, but in most cases, standard CP-ECE goes up. This indicates that training with more data does not necessarily improve the CP calibration.

**Noisy Environment.** Fig. 3 shows the CP reliability diagrams when input images are perturbed with Gaussian noise. Both ViT models are not as robust to Gaussian noise as the ResNet model, but the ResNet model has the worse standard CP-ECE compared with the other two. In particular, Fig. 3.1 shows that when there is more noise, $\tau$ will decrease, indicating that the probability shape within the prediction set gets more peaked. This finding is validated by the result in Fig. 6.a, where the entropy of $\tilde{p}$ within a prediction set when $\sigma = 0.8$ is smaller than that when test images are clean. However, there is an opposite trend in the entropy of the original probability $p$ as shown in Fig. 6.b. The CP reliability diagrams when Gaussian noise is added to the embeddings or there are typos in the text when using GPT-2 are shown in Fig. 5. The $\tau$ of clean input is slightly higher than that when there is input noise, but the change of $\tau$'s is minor in GPT-2 compared with that in vision models.

### 5.4 Performance of CPAC

The previous subsection shows that calibration is an independent dimension of CP and needs to be optimized. We present our empirical results on ImageNet and Topic Classification in this subsection, as the calibration error on CIFAR100 is not high, we focus on calibrating CP.

Tab. 1 shows the result of testing ViT-L and ViT-B when there are input noise for both calibration and test set. Tab. 2 shows the result of GPT2 on the topic classification task. On almost all the settings, CPAC reduces the Uniform CP-ECE without sacrificing Std CP-ECE or even improve it, and meanwhile maintains the accuracy and decreases the PSS. The

|  | Power | Exponential | Logarithmic |
|---|---|---|---|
| Uniform | 0.29(0.03) | 2.73(0.12) | 2.04(0.07) |
| Multinomial | 6.99(0.24) | 17.23(0.20) | 5.95(0.22) |
| Top-1 | 11.02(0.30) | 18.40(0.37) | 10.41(0.26) |

Table 3: Curve fitting error (Uni. CP-ECE) of ViT-L on ImageNet using different sampling strategies.

decreased PSS can be attributed to the CPAC on samples with low PSS. We also observe that CPAC mainly reduces the Uni. CP-ECE, especially when there are many classes, i.e., on ImageNet. This is due to the fact that the loss of high-PSS samples in CPAC is often larger than low-PSS samples, so CPAC tends to focus on high-PSS samples and leads to low curve fitting error. To compare the PSS when the coverage is fixed, we select the non-conformity score threshold so that the coverage is controlled to be the 90% and report the result in Appendix C. When the coverage is fixed, our method enlarges the PSS compared with the baseline (split CP with Platt scaling). However, the coverage control experiment is not doable in practice as the test set is unknown. The increased PSS won't diminish our major contributions, i.e., the concept and method of PSS calibration, as the table still shows the improvement of calibration error when using CPAC.

## 6 Conclusion

This work presents a systematic research into the uncertainty calibration in CP for classification, where the uncertainty is measured by the prediction set size. We first give a definition and metrics for the calibration of CP, then propose a target calibration function for PSS which is validated by both empirical results and our theoretical analysis. Finally, we propose a bi-level optimization algorithm that performs CP-aware calibration, and show its effectiveness on three classification tasks with state-of-the-art models. This work will inspire future research into the uncertainty calibration of CP, which is largely neglected by the community. One weakness of this work is that the convergence and generalization of the bi-level optimization problem are only validated empirically but not analyzed in theory, which will be addressed by our future work.

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

## A  PROOF

*Proof of Theorem 4.2.* By the definition of the Dirichlet distribution, for each coordinate $j$ we have

$$\mathbb{E}[p_j] \;=\; \frac{\alpha_j}{\alpha_0} \;=\; \frac{\alpha_0\, a_j}{\alpha_0} \;=\; a_j, \quad \text{and similarly} \quad \mathbb{E}[q_j] \;=\; \frac{\beta_j}{\beta_0} \;=\; a_j.$$

Since $p$ and $q$ are independent,

$$\mathbb{E}[\,p_j\, q_j\,] \;=\; \mathbb{E}[\,p_j\,]\, \mathbb{E}[\,q_j\,] \;=\; a_j^2.$$

Summing over $j = 1, \ldots, K$ gives

$$\mathbb{E}[p \cdot q] \;=\; \mathbb{E}\Big[\sum_{j=1}^{K} p_j\, q_j\Big] \;=\; \sum_{j=1}^{K} \mathbb{E}[\,p_j\, q_j\,] \;=\; \sum_{j=1}^{K} a_j^2,$$

proving the exact mean. The bounds follow from $\sum_{j=1}^{K} a_j = 1$ and the fact that $a_j \geq 0$, and the power-law exponent $\tau$ is obtained by taking the negative logarithm base $K$:

$$\tau = -\frac{\ln\left(\sum_{j=1}^{K} a_j^2\right)}{\ln K} \quad \Longrightarrow \quad \sum_{j=1}^{K} a_j^2 = K^{-\tau}.$$

Thus the theorem is established. $\qquad\square$

**Theorem A.1** (Expected accuracy under heterogeneous logistic–normal). *Let $Z_j \sim \mathcal{N}(0, \sigma_j^2)$ be independent Gaussian logits with $\sigma_1^2 \geq \sigma_2^2 \geq \cdots \geq \sigma_k^2 > 0$. If the variances satisfy the Lindeberg–Feller bounds*

$$\sigma_1^2 \leq C < \infty, \qquad \sum_{j=1}^{k} \sigma_j^2 = O(k), \tag{A.1}$$

*then for every fixed exponent $t > 1$*

$$Acc_t \;=\; \frac{C_{t,hetero}}{k} + O\big(k^{-3/2}\big) \qquad (k \to \infty),$$

*where*

$$C_{t,hetero} := \frac{\frac{1}{k} \sum_{j=1}^{k} \mu_{t,j}}{\left(\frac{1}{k} \sum_{j=1}^{k} \mu_{t-1,j}\right)\left(\frac{1}{k} \sum_{j=1}^{k} \mu_{1,j}\right)}, \quad \mu_{r,j} = \exp\!\Big(\tfrac{1}{2} r^2 \sigma_j^2\Big).$$

The expectation of sampling accuracy with temperature $t$ is defined, as in Equation (7) of the main paper, by

$$Acc_t = \mathbb{E}\left[\frac{\sum_{j=1}^{k} P_j^t}{\sum_{j=1}^{k} P_j^{t-1}}\right], \quad P_j = \frac{T_j}{\sum_{\ell=1}^{k} T_\ell}, \quad T_j = e^{Z_j}.$$

**Assumptions and notations.**

Latent logits: $\qquad\qquad Z_j \sim \mathcal{N}(0, \sigma_j^2), \quad 1 \leq j \leq k, \qquad \sigma_1^2 \geq \sigma_2^2 \geq \cdots \geq \sigma_k^2 > 0.$

Log–normals: $\qquad\qquad T_j := e^{Z_j}, \qquad \mu_{r,j} := \mathbb{E}[T_j^r] = \exp(\tfrac{1}{2} r^2 \sigma_j^2),$

$$\qquad\qquad\qquad\qquad\quad \varsigma_{r,j}^2 := \mathrm{Var}(T_j^r) = \big(e^{r^2 \sigma_j^2} - 1\big)\mu_{r,j}^2.$$

Softmax probabilities: $\quad P_j = \frac{T_j}{T_\Sigma}, \qquad T_\Sigma := \sum_{j=1}^{k} T_j.$

Power sums: $\qquad\qquad N_k := \sum_{j=1}^{k} T_j^t, \qquad D_k := \sum_{j=1}^{k} T_j^{t-1}.$

Deterministic means: $\quad \bar{\mu}_t := \sum_{j=1}^{k} \mu_{t,j}, \quad \bar{\mu}_{t-1} := \sum_{j=1}^{k} \mu_{t-1,j}, \quad \bar{\mu}_1 := \sum_{j=1}^{k} \mu_{1,j}.$

**Law of Large Numbers (Lindeberg–Feller).** Fix $r \in [2, t]$. Define the centred summand $\xi_{k,j}^{(r)} := T_j^r - \mu_{r,j}$ and total variance $V_{k,r}^2 := \sum_{j=1}^k \varsigma_{r,j}^2$. If

$$\max_{1 \le j \le k} \varsigma_{r,j}^2 = O(1) \implies V_{k,r}^2 \asymp k,$$

and the Lindeberg condition holds for every $\varepsilon > 0$,

$$\frac{1}{V_{k,r}^2} \sum_{j=1}^k \mathbb{E}\big[\xi_{k,j}^{(r)2} \mathbf{1}\{|\xi_{k,j}^{(r)}| > \varepsilon V_{k,r}\}\big] \xrightarrow[k \to \infty]{} 0,$$

then, since log–normals have super–polynomially decaying tails,

$$\frac{1}{\sqrt{V_{k,r}^2}} \sum_{j=1}^k \xi_{k,j}^{(r)} \Rightarrow N(0,1) \implies \frac{1}{k} \sum_{j=1}^k \xi_{k,j}^{(r)} = O_p(k^{-1/2}).$$

Hence

$$\frac{N_k - \bar{\mu}_t}{k}, \qquad \frac{D_k - \bar{\mu}_{t-1}}{k}, \qquad \frac{T_\Sigma - \bar{\mu}_1}{k} = O_p\big(k^{-1/2}\big).$$

A sufficient explicit condition is again (A.1). For descending variances $\sigma_j^2 = \sigma_1^2 j^{-\beta}$ with any $\beta > 0$, A.1 is satisfied.

**Fraction expansion with heterogeneous means.** Let

$$X_k = N_k - \bar{\mu}_t, \qquad Y_k = D_k - \bar{\mu}_{t-1}, \qquad Z_k = T_\Sigma - \bar{\mu}_1.$$

Then

$$\frac{N_k}{D_k T_\Sigma} = \frac{\bar{\mu}_t}{\bar{\mu}_{t-1} \bar{\mu}_1} \cdot \frac{1 + X_k/\bar{\mu}_t}{(1 + Y_k/\bar{\mu}_{t-1})(1 + Z_k/\bar{\mu}_1)}.$$

A second–order Taylor expansion yields

$$\frac{N_k}{D_k T_\Sigma} = \frac{\bar{\mu}_t}{\bar{\mu}_{t-1} \bar{\mu}_1} \Big[1 + O_p(k^{-1/2})\Big].$$

**Expectation and scaling law.** Taking expectations cancels linear terms:

$$Acc_t = \frac{\bar{\mu}_t}{\bar{\mu}_{t-1} \bar{\mu}_1} \Big[1 + O\big(k^{-1/2}\big)\Big].$$

Since $\bar{\mu}_r = \sum_{j=1}^k \mu_{r,j} = k\hat{\mu}_r$, the ratio of averages is $\Theta(1)$ and

$$Acc_t = \frac{k\hat{\mu}_t}{(k\hat{\mu}_{t-1})(k\hat{\mu}_1)} + O(k^{-3/2}) = \frac{C_{t,\text{hetero}}}{k} + O(k^{-3/2}),$$

with

$$C_{t,\text{hetero}} := \frac{\frac{1}{k} \sum_{j=1}^k \mu_{t,j}}{\big(\frac{1}{k} \sum_{j=1}^k \mu_{t-1,j}\big)\big(\frac{1}{k} \sum_{j=1}^k \mu_{1,j}\big)}.$$

# B  ADAPTIVE PREDICTION SETS (ROMANO ET AL., 2020)

Here is a description of the adaptive prediction sets (APS) method used in our paper. Suppose we have the prediction distribution $\boldsymbol{p}(x) = f_\theta(x)$ and order this probability vector with the descending order $p_{(1)}(x) \ge p_{(2)}(x) \ge \ldots \ge p_{(K)}(x)$. The generalized conditional quantile function is defined as,

$$Q(x; p, \nu) = \min\{k \in \{1, \ldots, K\} : p_{(1)}(x) + p_{(2)}(x) + \ldots + p_{(k)}(x) \ge \nu\}, \qquad (10)$$

which produces the class index with the generalized quantile $\nu \in [0, 1]$. The function $\mathcal{S}$ can be defined as

$$\mathcal{S}(x, u; p, \nu) = \begin{cases} \text{`}y\text{' indices of the } Q(x; p, \nu) - 1 \text{ largest } p_y(x), & \text{if } u \le U(x; p, \nu), \\ \text{`}y\text{' indices of the } Q(x; p, \nu) \text{ largest } p_y(x), & \text{otherwise,} \end{cases} \qquad (11)$$

where

$$U(x; p, \nu) = \frac{1}{p_{(Q(x;p,\nu))}(x)} \left[ \sum_{k=1}^{Q(x;p,\nu)} p_{(k)}(x) - \nu \right].$$

It has input $x, u \in [0, 1]$, $\pi$, and $\nu$ which can be seen as a generalized inverse of Equation 10.

On the calibration set $\mathcal{D}_{cal}$, we compute a generalized inverse quantile conformity score using the following function,

$$E(x, y, u; p) = \min \{\nu \in [0, 1] : y \in \mathcal{S}(x, u; p, \nu)\}, \tag{12}$$

which is the smallest quantile to ensure that the ground-truth class is contained in the prediction set $\mathcal{S}(x, u; p, \nu)$. With the conformity scores on calibration set $\{E_i\}_{i=1}^{N_{cal}}$, we compute the $\lceil(1 - \alpha)(1 + N_{cal})\rceil$th largest value in the score set as $\hat{\nu}_{cal}$. During inference, the prediction set is generated with $\mathcal{S}(\boldsymbol{x}^*, u; p^*, \hat{\nu}_{cal})$ for a test sample $\boldsymbol{x}^*$.

## C  MORE EXPERIMENT

Fig. 7 and 8 shows the reliability diagrams of PS and CPAC on ImageNet and ViT-L when uniform CP-ECE is used as the metric. The calibration error of CPAC is qualitatively better than that of PS. Similarly, we visualize the uniform CP-ECE comparison of PS and CPAC for ViT-B on ImageNet in Fig. 11 and 12. The results of standard CP-ECE are shown in Fig. 9 and 10 for ViT-L and Fig. 13 and 14 for ViT-B. Finally, we report the result of using PS and CPAC on ImageNet in Tab. 5 when ResNet101 is used, which demonstrates the strength of CPAC in reducing uniform CP-ECE.

## D  THE USE OF LLM

The use of LLMs is restricted to language refinement, including grammar correction, sentence rephrasing, and improving the clarity of writing. No LLMs were used to generate research ideas, design methodology, conduct experiments, or create results. All technical contributions, implementations, and analyses presented in this paper are solely the work of the authors.

|  |  | Std CP-ECE | Uni CP-ECE | Acc. | Cov. | PSS |
|---|---|---|---|---|---|---|
| Clean | PS | 8.68(0.01) | 7.34(0.71) | 80.35(0.09) | 90.00(0.01) | 6.20(0.10) |
|  | CPAC | 7.93(0.14) | 6.37(0.52) | 77.68(0.14) | 90.00(0.02) | 7.81(0.29) |
| Norm-0.1 | PS | 9.09(0.07) | 7.85(0.59) | 78.67(0.06) | 90.00(0.01) | 7.93(0.23) |
|  | CPAC | 8.46(0.09) | 6.25(0.28) | 78.67(0.06) | 90.00(0.01) | 7.13(0.18) |
| Norm-0.2 | PS | 9.85(0.12) | 8.85(0.37) | 74.13(0.05) | 89.99(0.04) | 14.82(0.15) |
|  | CPAC | 8.37(0.09) | 7.78(0.61) | 70.58(0.41) | 90.00(0.03) | 18.17(0.97) |
| Norm-0.4 | PS | 10.74(0.09) | 6.72(0.21) | 57.19(0.16) | 90.00(0.03) | 62.37(0.49) |
|  | CPAC | 7.58(0.33) | 7.68(0.44) | 51.97(0.27) | 89.97(0.05) | 93.87(2.89) |
| Norm-0.8 | PS | 5.02(0.12) | 7.23(0.21) | 12.30(0.06) | 90.01(0.02) | 429.82(3.04) |
|  | CPAC | 4.37(0.12) | 4.90(0.11) | 12.50(0.21) | 89.97(0.04) | 456.89(3.74) |
| Blur-3 | PS | 9.27(0.13) | 9.60(0.36) | 77.06(0.10) | 90.00(0.02) | 10.52(0.13) |
|  | CPAC | 8.46(0.14) | 7.24(0.68) | 75.19(0.32) | 90.02(0.01) | 11.76(0.44) |
| Blur-5 | PS | 9.81(0.06) | 8.91(0.45) | 75.24(0.09) | 90.00(0.03) | 13.39(0.14) |
|  | CPAC | 8.73(0.19) | 7.51(0.23) | 73.09(0.10) | 90.00(0.02) | 15.48(0.46) |
| Blur-7 | PS | 9.98(0.04) | 8.45(0.39) | 74.39(0.06) | 89.99(0.02) | 15.08(0.22) |
|  | CPAC | 8.65(0.19) | 7.93(0.26) | 71.12(0.20) | 89.98(0.03) | 19.14(0.84) |
| Drop-1 | PS | 10.93(0.07) | 8.67(0.39) | 71.04(0.07) | 90.00(0.05) | 21.45(0.34) |
|  | CPAC | 8.96(0.11) | 8.52(0.52) | 68.26(0.14) | 89.99(0.02) | 26.39(0.70) |
| Drop-3 | PS | 11.42(0.04) | 6.52(0.17) | 54.91(0.11) | 90.01(0.02) | 72.49(0.24) |
|  | CPAC | 8.22(0.39) | 7.60(0.47) | 52.37(0.76) | 90.00(0.03) | 89.10(5.19) |
| Drop-5 | PS | 8.95(0.14) | 5.00(0.10) | 35.38(0.07) | 90.00(0.05) | 171.88(1.52) |
|  | CPAC | 6.63(0.53) | 5.52(0.27) | 32.77(0.53) | 90.02(0.08) | 241.31(10.71) |
| Drop-7 | PS | 5.09(0.08) | 7.23(0.10) | 11.97(0.09) | 90.00(0.01) | 424.25(0.65) |
|  | CPAC | 4.72(0.17) | 5.01(0.19) | 12.73(0.22) | 90.02(0.06) | 450.48(3.68) |

Table 4: Performance metrics of ViT-Base on ImageNet when the coverage is controlled to be 90%.

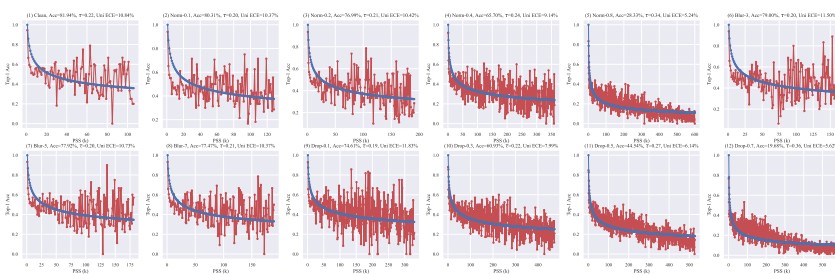

Figure 7: Reliability diagrams of ViT-L on ImageNet under different types of noise with uniform CP-ECE.

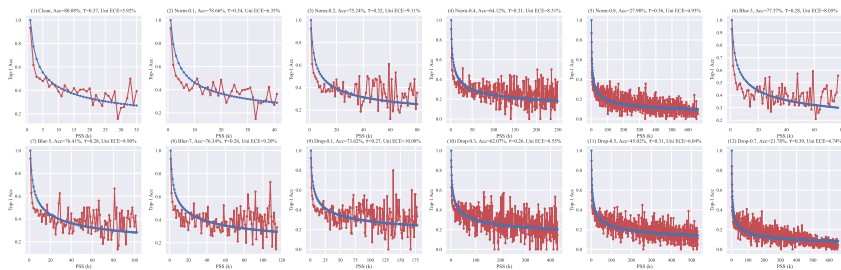

Figure 8: Reliability diagrams of ViT-L on ImageNet under different types of noise with uniform CP-ECE after CPAC.

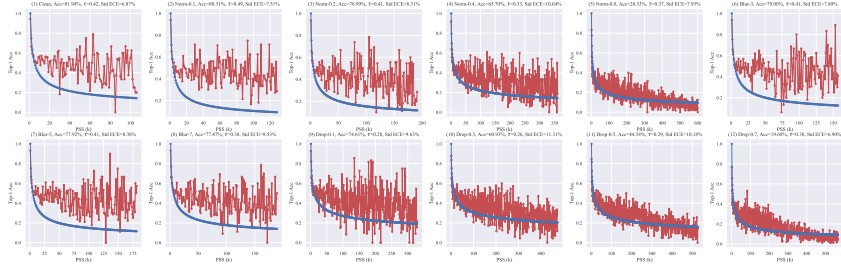

Figure 9: Reliability diagrams of ViT-L on ImageNet under different types of noise with standard CP-ECE.

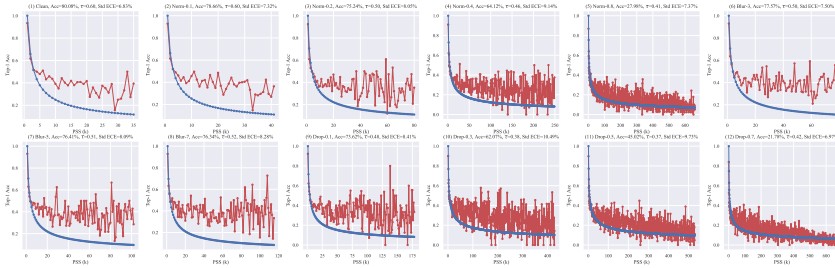

Figure 10: Reliability diagrams of ViT-L on ImageNet under different types of noise with standard CP-ECE after CPAC.

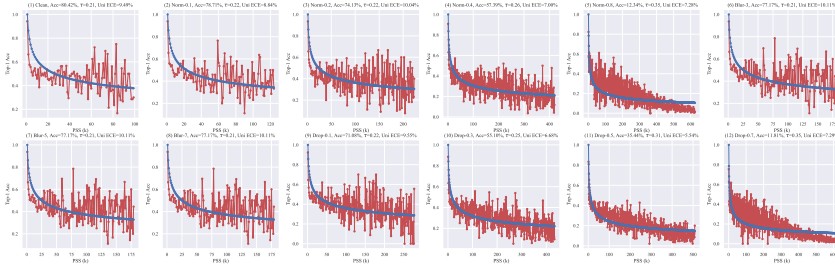

Figure 11: Reliability diagrams of ViT-B on ImageNet under different types of noise with uniform CP-ECE.

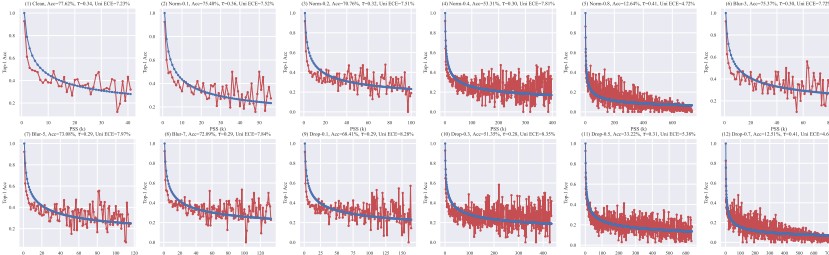

Figure 12: Reliability diagrams of ViT-B on ImageNet under different types of noise with uniform CP-ECE after CPAC.

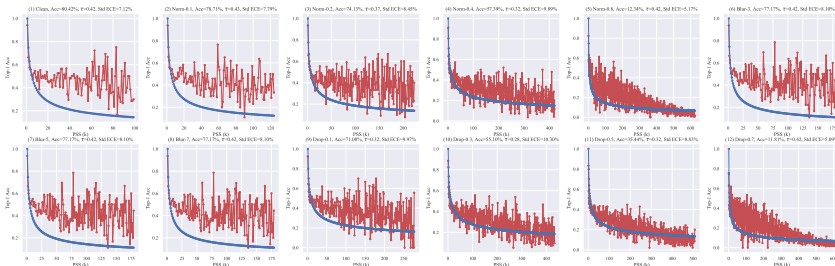

Figure 13: Reliability diagrams of ViT-B on ImageNet under different types of noise with standard CP-ECE.

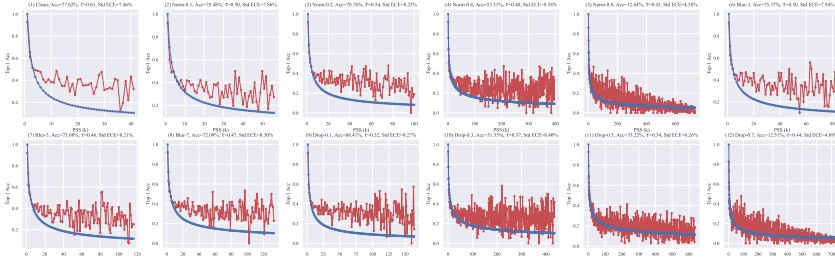

Figure 14: Reliability diagrams of ViT-B on ImageNet under different types of noise with standard CP-ECE after CPAC.

| | | Std CP-ECE | Uni CP-ECE | Acc. | Cov. | PSS |
|---|---|---|---|---|---|---|
| | PS | 7.01(0.14) | 11.33(0.92) | 81.93(0.13) | 93.46(0.29) | 16.04(1.59) |
| Clean | PS-Full | 9.20(0.15) | 7.85(0.23) | 81.93(0.13) | 94.77(0.21) | 29.53(1.46) |
| | CPAC | 9.16(0.13) | 7.82(0.25) | 81.93(0.13) | 94.75(0.18) | 29.57(1.18) |
| | PS | 7.44(0.17) | 10.87(1.41) | 80.76(0.12) | 93.36(0.33) | 17.73(2.05) |
| Norm-0.1 | PS-Full | 9.12(0.19) | 8.07(0.20) | 80.76(0.12) | 94.65(0.26) | 29.99(1.62) |
| | CPAC | 9.11(0.15) | 7.91(0.24) | 80.76(0.12) | 94.63(0.20) | 30.18(1.38) |
| | PS | 7.90(0.14) | 11.24(0.72) | 79.18(0.13) | 93.13(0.38) | 19.89(2.27) |
| Norm-0.2 | PS-Full | 9.47(0.13) | 7.66(0.19) | 79.18(0.13) | 94.29(0.20) | 30.22(1.33) |
| | CPAC | 9.40(0.19) | 7.67(0.44) | 79.20(0.14) | 94.39(0.23) | 30.92(1.42) |
| | PS | 8.74(0.17) | 9.58(0.38) | 74.40(0.07) | 92.63(0.31) | 28.04(2.16) |
| Norm-0.4 | PS-Full | 9.41(0.14) | 7.31(0.21) | 74.40(0.07) | 93.76(0.24) | 35.76(1.57) |
| | CPAC | 9.39(0.18) | 7.08(0.21) | 74.40(0.07) | 93.83(0.34) | 36.53(2.33) |
| | PS | 9.23(0.06) | 6.65(0.16) | 58.57(0.11) | 91.53(0.25) | 67.89(2.20) |
| Norm-0.8 | PS-Full | 8.44(0.10) | 5.51(0.11) | 58.57(0.11) | 91.97(0.11) | 63.14(0.94) |
| | CPAC | 8.36(0.07) | 5.33(0.18) | 58.63(0.09) | 91.94(0.20) | 62.27(1.74) |
| | PS | 7.47(0.15) | 10.71(0.36) | 79.88(0.14) | 93.31(0.28) | 18.56(1.43) |
| Blur-3 | PS-Full | 9.31(0.18) | 7.99(0.13) | 79.88(0.14) | 94.60(0.18) | 30.69(1.47) |
| | CPAC | 9.29(0.20) | 7.82(0.43) | 79.88(0.16) | 94.59(0.10) | 30.64(0.84) |
| | PS | 8.04(0.18) | 10.66(0.35) | 78.17(0.11) | 93.16(0.25) | 22.38(1.51) |
| Blur-5 | PS-Full | 9.41(0.14) | 7.42(0.36) | 78.17(0.11) | 94.44(0.22) | 33.23(1.57) |
| | CPAC | 9.39(0.15) | 7.42(0.40) | 78.17(0.11) | 94.44(0.15) | 33.15(1.09) |
| | PS | 8.26(0.21) | 10.23(0.38) | 77.45(0.12) | 93.06(0.25) | 23.94(1.53) |
| Blur-7 | PS-Full | 9.38(0.20) | 7.40(0.28) | 77.45(0.12) | 94.34(0.22) | 34.34(1.59) |
| | CPAC | 9.35(0.17) | 7.27(0.17) | 77.45(0.12) | 94.34(0.17) | 34.14(1.25) |
| | PS | 10.92(0.17) | 8.48(0.23) | 67.03(0.13) | 91.81(0.29) | 55.23(2.38) |
| Drop-1 | PS-Full | 10.36(0.23) | 6.63(0.30) | 67.03(0.13) | 92.59(0.25) | 55.07(1.98) |
| | CPAC | 9.97(0.24) | 6.40(0.19) | 67.37(0.13) | 92.79(0.24) | 54.24(1.85) |
| | PS | 10.72(0.31) | 6.36(0.18) | 52.05(0.12) | 90.64(0.34) | 108.84(4.37) |
| Drop-3 | PS-Full | 9.87(0.31) | 5.69(0.25) | 52.05(0.12) | 90.88(0.35) | 98.10(3.90) |
| | CPAC | 9.10(0.23) | 5.42(0.28) | 53.11(0.12) | 91.11(0.36) | 87.01(3.53) |
| | PS | 8.25(0.20) | 5.03(0.17) | 37.59(0.07) | 90.29(0.37) | 182.23(6.52) |
| Drop-5 | PS-Full | 8.02(0.12) | 4.89(0.20) | 37.59(0.07) | 90.29(0.33) | 176.38(5.28) |
| | CPAC | 7.31(0.13) | 4.71(0.09) | 39.03(0.10) | 90.35(0.30) | 142.69(3.28) |
| | PS | 4.75(0.14) | 5.55(0.07) | 19.41(0.06) | 89.94(0.27) | 328.26(6.81) |
| Drop-7 | PS-Full | 5.82(0.17) | 5.48(0.08) | 19.41(0.06) | 90.00(0.31) | 359.11(7.00) |
| | CPAC | 4.77(0.26) | 5.09(0.28) | 21.14(0.21) | 89.47(0.28) | 300.94(5.65) |

Table 5: Result of ResNet101 on ImageNet-1k.

