# OpenReview forum: "Is the Prediction Set Size Well-Calibrated? A Closer Look at Uncertainty in Conformal Prediction"
_ICLR.cc/2026/Conference — ICLR 2026 Conference Withdrawn Submission_

### Official Review · Reviewer_sbHn · 2025-10-27

**Soundness:** 3
**Presentation:** 3
**Contribution:** 3
**Rating:** 6
**Confidence:** 2

**Summary:**

The authors identify that the relationship between PSS and model reliability has not been systematically studied in classification tasks. To address this, the paper:
1. Defines a notion of CP calibration by linking PSS to expected accuracy using a multinomial sampling strategy.
2. Proposes a calibration target function, theoretically motivated by Dirichlet distribution assumptions, to model the expected relationship between PSS and accuracy.
3. Introduces CP-Aware Calibration (CPAC) to improve this calibration while maintaining coverage guarantees.

**Strengths:**

1. The paper addresses an important aspect of CP: whether uncertainty conveyed by PSS aligns with correctness. This extends calibration analysis from probabilistic outputs to set-valued predictions, a meaningful and clear contribution.

2. The definition of CP calibration and the derivation of a target calibration function from Dirichlet assumptions look sound.

3. The experiments span two datasets (CIFAR-100 and ImageNet-1k) and architectures (ResNet, ViT, GPT-2). The inclusion of noisy environments (Gaussian, blur, dropout, typos) strengthens the generality of the findings. The authors also include ablation studies (different target functions, pretraining/random initialization, and sampling temperatures).

**Weaknesses:**

1. Does CPAC formally preserve the (1–$\alpha$) coverage guarantee under its bi-level optimization? If not, how can we quantify or bound potential deviations?


2. While CPAC is compared to Platt scaling and standard temperature scaling within APS, it would be valuable to see comparisons against *post-hoc recalibration* methods (e.g., histogram binning, isotonic regression) or *recent conformal calibration* methods (e.g., random set calibration, Bayesian CP).

3. The reported experimental results are dense and broad, while the **analysis** on experiments is occasionally "superficial". For instance:

3a. Why CPAC sometimes reduces coverage slightly is not well explained.

3b. Is there a theoretical or empirical analysis of the calibration-efficiency frontier? (Similar to 3a)

3c. The effect of hyperparameters ($\lambda$, $\tau$, learning rate) could be more explored.


4. The assumption that *predictive probabilities and sampling distributions follow a shared Dirichlet form* is strong and **not empirically validated**. Alternative formulations (e.g., logistic-normal) are briefly mentioned but not deeply compared.
Can the authors provide empirical evidence supporting the Dirichlet assumption on predictive probabilities?


5. Concernig the presentation:

5a. The texts (such as sub-titles, x-axis, and y-axis) on Figures 2-6 are too small to distinguish, leading to poor presentation.

5b. Tables 1-2 are too dense.

**Questions:**

See above

---

### Official Review · Reviewer_YDJk · 2025-10-29

**Soundness:** 3
**Presentation:** 2
**Contribution:** 3
**Rating:** 4
**Confidence:** 3

**Summary:**

This paper investigates the calibration of prediction set size (PSS) in conformal prediction (CP), questioning whether smaller prediction sets truly correspond to higher predictive accuracy. The authors first formalize the notion of conformal calibration-a monotonic relationship between the prediction set size and expected accuracy-and propose CP-Aware Calibration (CPAC), a bi-level optimization procedure that adjusts model logits to better align the PSS-accuracy relationship with a theoretically motivated function
$f(k)=1/k^{\tau}$. Empirical results across image and text classification tasks (ResNet, ViT, GPT-2) show that CPAC improves calibration metrics (CP-ECE and Uniform CP-ECE) while maintaining nominal coverage.

**Strengths:**

1.	Practical algorithm (CPAC): The proposed bi-level calibration method is simple, differentiable, and can be used as a plug-in for existing models.
2.	Comprehensive experiments: Evaluation spans diverse architectures (ResNet, ViT, GPT-2) and domains (image and text), demonstrating robustness.

**Weaknesses:**

1.	Dependence on APS score:
All experiments are based on the Adaptive Prediction Sets (APS) scoring function. No experiments are provided for RAPS and other score functions, which can exhibit different prediction set behaviors. This limits the generality of the proposed CPAC method.
2.	Dirichlet assumption and lack of theoretical grounding: The derivation of $f(k)=1/k^{\tau}$  assumes a Dirichlet distribution for softmax outputs, which is rarely realistic for modern neural networks. The sensitivity of CPAC to this assumption is not analyzed.
3.	Unaddressed Negative Trade-offs (Efficiency): A crucial experiment is relegated to Appendix C (Table 4), which shows results when the coverage is controlled to be 90%. This table reveals that the proposed CPAC method consistently produces larger (i.e., worse) Prediction Set Sizes (PSS) than the baseline (e.g., ViT-Base "Clean" PSS increases from 6.20 to 7.81).

**Questions:**

1.	On the $\tau$ Parameter: Please clarify precisely how the target calibration parameter $\tau$ is determined for (a) the CPAC algorithm's objective function (Algorithm 1) and (b) the final ECE evaluation (e.g., in Table 1). How do you justify using a test-set grid search to find the "optimal $\tau$" for your evaluation metric?
2.	How do you interpret the finding in Table 4  that CPAC increases the PSS when coverage is fixed? Does this not imply a fundamental trade-off between your definition of calibration and the primary goal of CP, which is to provide the smallest possible set while guaranteeing coverage?

---

### Official Review · Reviewer_HbKC · 2025-10-30

**Soundness:** 2
**Presentation:** 2
**Contribution:** 2
**Rating:** 2
**Confidence:** 4

**Summary:**

The paper studies the relation between the model's accuracy and the prediction set size (PSS) of conformal prediction applied to classification (empirically, it only considers the APS method). They refer to the PSS as calibrated if it monotonically decreases with the PSS and finds that it is not always the case. Metrics and a method that improves them are proposed.

**Strengths:**

The research direction is novel and interesting, but I find some issues with the current state of the paper.

**Weaknesses:**

1.
If both calibration and CP are performed using the same calibration set, the exchangeability condition between calibration and test samples, which is required for proving Eq. (1), is not ensured.

2.
The PSS calibration criterion (in particular the expression for accuracy) is somewhat heuristic.
There is no "true" monotonically decreasing function f(|S|) that all users are expected to prefer.

3.
The discussion below Eq. 9 is not clear enough.
Explain more the case of samples with PSS=1, and also explain what happens for samples with PSS=0, which can happen in some CP methods.
At some later point in the paper, the authors state that they prevent the case of PSS=0, but this degrades the performance of CP methods, as the option of empty PS allows them to reach the desired coverage level with smaller average PSS, and empty PSS serves as an indication to the user that a sample is so hard that the model cannot suggest a candidate for it (a "reject" option).

4.
You consider only the APS score.
Yet, it is known that APS performs poorly in terms of PSS (prediction set size), and the sets even become larger after performing calibration (Xi et al., 2024), (Dabah & Tirer, 2024).
You should repeat the experiments also with other popular CP scoring methods, such as RAPS and especially LAC/THR (one minus the softmax entry), which have better PSS.

5.
The proposed calibration affects the top-1 accuracy (W,b can change the class ranking) and degrades the accuracy of the model, as shown in Table 1.

6.
What value of alpha do you use? (I suggest examining more than one value).
I assume that you used 0.1, and it is not clear why the target coverage of 1-\alpha is not obtained more precisely for clean test data, contrary to existing works.
Perhaps this relates to negative effect on exchangeability since you perform both CP and calibration on the same set.

7.
Regarding your statement: "as we only need to cover (1 − α) of all samples in CP and we choose to optimize those low-PSS samples", I do not think that this is justified, as you do not know in which 1-\alpha of the samples coverage holds.

8.
You state that you use: "PS to denote the standard confidence calibration method in APS". However, confidence calibration has been shown to have negative effect on the PSS of APS. Thus, you should compare also to APS without confidence calibration.
In addition, "PS-Full" is not defined.

9.
The metric "Uni CP-ECE", where all the bins (PSS values) have fixed weight contrary to "Std CP-ECE" that matches the empirical distribution, is not well justified. Why don't you use adaptive binning?
The proposed CPAC does not seem to yield significant improvements on Std CP-ECE, at least in the "clean" cases, which are more natural, in Table 1. (The "noisy cases" are less natural, as the model was trained on clean data and no domain adaptation is used.)

10.
In Table 4 in the appendix, the coverage level is aligned with 1-\alpha, which is the common observation for CP methods on standard benchmarks (not in a "control experiment"). In this case, the proposed CPAC both degrades accuracy and increases PSS compared to the baseline.

**Questions:**

I stated the questions above.

---

### Official Review · Reviewer_ivD8 · 2025-11-01

**Soundness:** 2
**Presentation:** 2
**Contribution:** 2
**Rating:** 2
**Confidence:** 3

**Summary:**

This paper investigates whether the prediction set size (PSS) in conformal prediction (CP) truly reflects predictive uncertainty in classification tasks. While CP guarantees coverage (the true label is contained in the prediction set with probability $1-a$), it is unclear if smaller sets consistently correspond to higher correctness. The authors formalize uncertainty calibration for CP, define metrics (CP-ECE, uniform CP-ECE), and propose a theoretical target function linking PSS and expected accuracy under Dirichlet assumptions. They further introduce CP-aware Calibration (CPAC), a bi-level optimization algorithm that adjusts model logits before conformalization to improve calibration. Experiments on CIFAR-100, ImageNet (ResNet, ViT models), and topic classification with GPT-2 demonstrate improved calibration (lower CP-ECE) without degrading accuracy or coverage.

**Strengths:**

- The paper tackles a previously underexplored question - whether prediction set size in CP reflects calibrated uncertainty - bridging a gap between coverage guarantees and reliability of uncertainty estimates.
- The proposed calibration target function $f(k)=1/k^\tau$ is motivated by both empirical observation and derivation under Dirichlet assumptions.
- The CPAC algorithm elegantly adapts bi-level optimization for CP calibration, analogous to Platt scaling but adapted to PSS.
- Evaluation across vision and language models, multiple perturbations and datasets provides strong empirical support.
- The study offers diagnostic insights (e.g., pre-trained models show weaker CP calibration; noise increases calibration errors).

**Weaknesses:**

- CPAC’s convergence and generalization are only empirically validated; no formal analysis is provided (acknowledged by authors).
- The bi-level optimization approach may be computationally heavy and lacks sensitivity analysis or ablation on hyperparameters (e.g. $\lambda, \tau, t$).
- The interpretation of the exponent parameter $\tau$ in the target function, while intuitive, could benefit from more rigorous justification.
- Some derivations (e.g., expected accuracy under multinomial sampling) could be made more transparent; reliance on Dirichlet assumptions might be restrictive.
- While CPAC improves calibration, it sometimes reduces accuracy slightly and increases PSS when coverage is fixed - this trade-off deserves more discussion.

**Questions:**

- How sensitive is CPAC to the choice of $\tau$ and the sampling temperature $t$? Could $\tau$ be learned rather than fixed via grid search?
- Can CPAC be extended to regression or structured prediction tasks?
- How does CPAC interact with existing conformal methods aimed at conditional coverage (e.g., Gibbs & Candes, 2023)?
- Would end-to-end fine-tuning (rather than post-hoc calibration) yield stronger alignment between PSS and accuracy?
- What is the computational overhead of CPAC relative to standard APS or Platt scaling?

---

### Note · Authors · 2025-11-12

I have read and agree with the venue's withdrawal policy on behalf of myself and my co-authors.